



# Ground-based remote sensing of aerosol properties using high resolution infrared emission and Lidar observations in the high Arctic

Denghui Ji[1], Mathias Palm[1], Christoph Ritter[2], Philipp Richter[1], Xiaoyu Sun[1], Matthias Buschmann[1], and Justus Notholt[1]

[1]Institute of Environmental Physics, University of Bremen, Otto-Hahn-Allee 1, 28359 Bremen, Germany
[2]Alfred Wegener Institute, Helmholtz Centre for Polar and Marine Research, Telegrafenberg A43, 14473 Potsdam, Germany

**Correspondence:** Denghui Ji (denghui_ji@iup.physik.uni-bremen.de)

**Abstract.** Arctic amplification, the phenomenon that the Arctic is warming faster than the global mean, is still not fully understood. The Transregional Collaborative Research Centre TR 172 – Arctic Amplification: Climate Relevant Atmospheric and Surface Processes $(AC)^3$ funded by the DFG (German research foundation) contribute towards this research topic. For the purpose of measuring aerosol components, a Fourier-Transform InfraRed spectrometer (FTS) for measuring down-welling

emission since 2019 and a Raman-Lidar are operated at the AWIPEV research base in Ny-Ålesund, Spitsbergen (79 °N, 12 °E). To do aerosol retrieval using measurements from the FTS, a retrieval algorithm based on Line-by-Line Radiative Transfer Model and DIScrete Ordinate Radiative Transfer model (LBLDIS), is modified for different aerosol types (dust, sea salt, black carbon, and sulfate), aerosol optical depth (AOD) and effective radius ($R_{eff}$). Using Lidar measurement, an aerosol and cloud classification method is developed for providing basic information about the distribution of aerosols or clouds in the atmosphere

and used as an indicator to do aerosols or clouds retrieval in FTS. Therefore, a two-instruments joint observation scheme is designed and is performing on the data measured from 2019 to present. In order to show this measurement technique in details, two case studies are selected, one is an aerosol-only case on the $10^{th}$ of June 2020 and the another is a cloud-only case on $11^{th}$ of June 2020. In the aerosol-only case, the retrieval results show that sulfate ($\tau_{900cm^{-1}}$=0.007 ± 0.0027) is the dominant aerosol during the whole day, followed by dust ($\tau_{900cm^{-1}}$=0.0039 ± 0.0029) and black carbon ($\tau_{900cm^{-1}}$=0.0017 ± 0.0007).

Sea salt ($\tau_{900cm^{-1}}$=0.0012 ± 0.0002) shows the lowest AOD value as its weakest emission ability in infrared waveband. Such proportions of sulfate, dust and BC also show good agreement with MERRA-2 reanalysis data. Besides, comparing with sun-photometer (AERONET), the daily variation of aerosol AOD retrieved from FTS is similar with that in sun-photometer. In the cloud-only case study, Lidar distinguishes the cloud signal from aerosols accurately, giving a very good information on the state of the atmosphere. For showing the importance of Lidar measurement in the retrieval of FTS, two versions of

retrieval algorithm, one for cloud retrieval and another for aerosols retrieval are applied for gaining cloud parameters and aerosol parameters respectively. The result shows that without information from Lidar measurement, the signal of cloud is misunderstood and retrieved as four aerosols in FTS, which indicates that the combination of both measurements is necessary and helpful in our aerosol retrieval.



# 1   Introduction

In the Arctic, near surface temperatures are rising much faster than those of the global mean (Wendisch et al., 2017). This phenomenon is called Arctic Amplification (Serreze and Barry, 2011; Wendisch et al., 2017; Previdi et al., 2021). In order to understand the causes and effects of the rapid warming in the Arctic, many studies focus on key processes contributing to Arctic amplification, like temperature feedback (Bony et al., 2006; Soden and Held, 2006), surface albedo feedback (Graversen et al., 2014) and cloud and water vapor feedback (Taylor et al., 2013; Philipp et al., 2020). The cooperative research programm (AC)[3] focuses on the Arctic Amplification[1].

Apart from the physical feedback processes, aerosol has a large impact on the Arctic environment (Abbatt et al., 2019; Schmale et al., 2021). Aerosol influences the Arctic climate by aerosol-cloud interactions (Fan et al., 2016) and aerosol-surface interactions (Donth et al., 2020). For example, Black Carbon (BC) deposits on snow and ice, lowering the surface albedo (Ming et al., 2009; Bond et al., 2013) and thus warming the surface. Dust, when present in layers over high albedo surfaces and/or deposited to the snow, will warm the atmosphere (Krinner et al., 2006); Sulfate, organic matter and sea salt may cool the Arctic by scattering light back to space and by modifying the microphysics of liquid clouds (Schmeisser et al., 2018). At cirrus temperatures, dust, ammonium sulfate and sea salt increase the cloud albedo by increasing ice crystal concentrations (Wagner et al., 2018).

In recent decades, there were mainly in-situ measurements of aerosols in the Arctic. Most reports show that the aerosol composition is changing. Koch et al. (2011) and Ren et al. (2020) find that sulfate and BC are decreasing compared to the last century. Several projects in (AC)[3] also focuses on BC concentration measurements (Kodros et al., 2018; Zanatta et al., 2018) and reveal the annual cycle of BC in the Arctic, higher in spring and lower in early summer (Schulz et al., 2019). Shaw (1995), Francis et al. (2018), Francis et al. (2019) find that dust can be transported over long distances into the Arctic and plays an important role in Arctic haze. In recent years, the area of open water becomes larger, and the sea surface temperature is increasing, which leads to a local increase of the emission of sea salt (Domine et al., 2004; Struthers et al., 2011; May et al., 2016). Thus, during the Arctic warming period, the proportions of different aerosols in the Arctic could also change.

There are several ways to measure the aerosol composition, such as remote sensing from satellite or ground-based instruments, in-situ measurement on the surface or from aircraft. Satellite instruments can provide measurements of large areas but are not very suitable in the Arctic due to the frequent existence of clouds and snow/ice on the surface, which make the measurements challenging (Lee et al., 2021). The in-situ measurements provide much more accurate measurements, but are often limited to the planetary boundary layer and a distinct position. The ground-based remote sensing method avoids the disturbance from the surface and has a similar viewing geometry as the satellite. Hence, a combination of different measurement methods is necessary to provide a complete picture of aerosols in the Arctic. In this paper, we focus on the ground-based remote sensing method in the thermal infrared to retrieve the aerosol components.

In section 2, the location of measurement site and the setup of two instruments is described. Section 3 presents the joint observation scheme using two instruments (Lidar KARL and the Fourier Transform spectrometer, NYAEM-FTS) and infor-

---

[1]www.ac3-tr.de





mation of the retrieval algorithm for NYAEM-FTS, including details of look-up tables of aerosol scattering properties. Section 4 shows the results of both KARL and NYAEM-FTS measurements in two case studies on aerosol-only event (10[th] June 2020) and cloud-only event (11[th] June 2020). The article ends with a summary and conclusion in section 5.

## 2 Location and Instruments Description

### 2.1 Site description

Ny-Ålesund (79 °N, 12 °E), Svalbard, is located in the North Atlantic atmospheric transport gateway to the Arctic. The AWIPEV[2] research base is part of the village Ny-Ålesund, jointly operated by the AWI Potsdam[3] and the IPEV institute[4].The AWI Potsdam operates an extensive suite of instruments, some of which are a very useful combination to the NYAEM-FTS (section 3.2), including an aerosol Lidar instrument (section 2.3) and a sun-photometer (appendix A).

### 2.2 NYAEM-FTS Measurement

The Fourier Transform spectrometer, NYAEM-FTS, for measuring down-welling emission in the thermal infrared was installed in summer 2019. The NYAEM-FTS consists of a Bruker Vertex 80 Fourier Transform Spectrometer, a SR800 blackbody an automatically operated mirror to select the radiation source and an automatically operated hutch with shields the instrument from the environment. It is situated in a temperature stabilized laboratory, at about 21 - 25 °C.

The Bruker Vertex 80 instrument is a table-top instrument. It is operated in zenith geometry with an adjustable field of view in the range of 3.3 mrad to 22 mrad. The beamsplitter is a KBr beam splitter and the detector is an extended MCT detector with the spectral range 400 - 2500 cm$^{-1}$ (4 - 25 µm). This instrument measures with a spectral resolution of 0.08 cm$^{-1}$ (08.2019 - 08.2020) and 0.3 cm$^{-1}$ (08.2020 to present). Spectra in different resolutions are all suitable for aerosol retrieval. The mirror selecting the emission source is the first optical part of the setup, and a total power calibration is performed to gain the radiance from spectra (Revercomb et al., 1988). This means three measurements are required to complete the observation of a spectrum, two measurements of the blackbody in hot and ambient temperature respectively, and one measurement pointing skywards (Rathke and Fischer, 2000; Turner, 2005; Richter et al., 2022). The SR800 blackbody is used as a blackbody radiator. It can be adjusted between 0 and 120 °C, holding the temperature within 0.1 K.

Figure 1 shows four different emission spectra measured by NYAEM-FTS in clear day, thick cloud, thin cloud and aerosol event respectively. The Planck function at 280 K is also presented in this figure. From Fig. 1, the intensity of thick cloud emission in infrared is high, close to the one calculated from Planck function. Compared with thick cloud, the atmospheric window between 800 - 1200 cm$^{-1}$ in the emission spectrum in clear day is obvious and the intensity of the spectral baseline is close to zero. Between the emission spectra of thick cloud and clear day in this window (800 - 1200 cm$^{-1}$), an aerosol and a thin cloud emission spectra are presented as well, showing the baseline, from which the aerosol information will be retrieved.

---

[2]www.awipev.eu

[3]Alfred Wegener Institut; www.awi.de

[4]Polar Institute Paul Emile Victor; www.ipev.fr





In general, it is easy to distinguish the difference between a spectrum of thick cloud with a spectrum in clear day, however, distinguishing aerosols from a thin cloud is difficult or impossible. Therefor more information from other instruments, e.g. Lidar measurement is used to distinguish days with clouds from days with aerosols present above the instrument.

## 2.3 The Raman-Lidar "KARL"

In Ny Ålesund, a Raman-Lidar "KARL" is operated to measure in 3 colors (355, 532, and 1064 nm) (Ritter et al., 2016). It is positioned about 10 meters away from NYAEM-FTS measurement. Aerosol backscatter coefficients (in all three colors), extinction coefficients (355 and 532 nm), and depolarization (355 and 532 nm) are measured.

For Lidar products, aerosol backscatter coefficient ($\beta^{aer}$), aerosol depolarization ($\delta^{aer}$) and color ratio ($CR$) are used for aerosol optical property analysis. According to Freudenthaler et al. (2009), the definitions of those quantities are given as
follows:

$$\delta^{aer}(\lambda) = \frac{\beta^{aer}_{\perp}(\lambda)}{\beta^{aer}_{\parallel}(\lambda)} \tag{1}$$

$\beta^{aer}_{\perp}(\lambda)$ and $\beta^{aer}_{\parallel}(\lambda)$ are the backscatter coefficients of the vertical and parallel polarized light, respectively. The depolarization depends on the particles' shape, e.g. spherical particles do not show any depolarization.

$$CR(\lambda_1, \lambda_2) = \frac{\beta^{aer}_{\lambda_1}}{\beta^{aer}_{\lambda_2}} \tag{2}$$

$\beta^{aer}_{\lambda}$ is the aerosol backscatter coefficient at wavelength $\lambda$.

The definitions of those parameters and more details are given in Freudenthaler et al. (2009) and Ritter et al. (2016). Based on that, Ritter et al. (2016) distinguished six conditions for the aerosol classification using those Lidar quantities (compare Tab. 1).

## 3 Methods and Data

### 3.1 Instruments Joint Observation Scheme

As we previously indicated, it is difficult to distinguish between thin clouds and aerosols only relying on the NYAEM-FTS instrument. To select the spectra in aerosol-only scenarios, the measurements of the KARL Lidar are used (compare Sec. 2.3).

First, the presence and distribution of clouds or aerosols are distinguished using the Lidar classification method (Sec. 2.3). The aerosol or cloud height can also be determined by Lidar and fixed in the retrieval algorithm for the FTS retrieval algorithm.
For cloud-only observations, the first version of retrieval algorithm, Total Cloud Water retrieval (TCWret-V1), will be used to do cloud parameters retrieval, which is described in Richter et al. (2022). For aerosol-only events, the modified version of TCWret-V2 will be adopted to do the aerosol components retrieval, which will be given in the following section. For complex situations of the simultaneous existence of clouds and aerosols, the concurrent FTS measurements will be excluded according to Lidar measurement. The flow diagram of instruments joint observation scheme could be found in Fig. 2a. Based on the





Lidar measurement, NYAEM-FTS will be used for cloud or aerosol retrieval using the corresponding databases for scattering coefficients.

      The flow diagram in TCWret-V2 is given in Fig. 2b. As shown in Fig. 2b, there are four inputs should be prepared for model simulation. First, the databases for scattering coefficients of different aerosol types are calculated using Mie code (Mishchenko et al., 1999) based on aerosol complex refractive index and aerosol size distribution. Second, the atmospheric state profile,

which includes temperature, humidity, and pressure (refered as THP), is obtained from ERA5 reanalysis data with a time resolution of 3 hours (Hersbach et al., 2018). The third input of the DIScrete Ordinate Radiative Transfer model (DISORT, Stamnes et al. (1988)) is the optical depth of gases in the whole atmosphere, which is calculated from the Line-by-Line Radiative Rransfer Model (LBLRTM, Clough et al. (2005)), using THP profile that we mentioned before. The last input for DISORT is the aerosol height information, which is provided by Lidar (Sec. 2.3). To obtain the temperature of the aerosol

layer, it is interpolated from ERA5 temperature data based on the height measured by Lidar. Furthermore, for all aerosol types, the apriori information of aerosol is fixed as AOD = 0.0001 and $R_{eff}$ = 0.35 μm. With the preparation of all input data, the model can simulate the spectrum and then use the retrieval algorithm to retrieve the aerosol parameters, trying to making the simulated one closer to the measured one. All of these processes will be explained in detail in the following section.

## 3.2    Retrieval Algorithm in NYAEM-FTS

The retrieval algorithms, TCWret-V1 and TCWret-V2, are based on TCWret developed by Richter et al. (2022). TCWret-V2 is modified for aerosol retrieval. The main difference between TCWret-V1 and TCWret-V2 is the scattering properties look-up table. Besides, the algorithm reliability, or how well the method can precisely retrieve aerosol information has been tested. In the Sec. 3.2.1, the aerosol scattering properties look-up tables and artificial spectra simulated using forward model are described in detail. The theory of retrieval algorithm can be found in App. B.

### 3.2.1    Aerosol Scattering Properties Look-up Tables

      Aerosol types can be categorized in several ways. In this study, sulfate, sea salt, dust, and BC are adopted in the retrieval algorithm. This is consistent with the reanalysis data, such as the MERRA-2 reanalysis data (Gelaro et al., 2017), which is convenient for data comparison. On the other hand, the complex refractive index database only covers the above aerosols in the infrared band. Furthermore, the residual term of spectral fitting is too small to consider other aerosol databases for

inversion. The complex imaginary refractive index of sulfate and dust are based on OPAC/GADA database, BC from Chang and Charalampopoulos (1990), and sea salt from Eldridge and Palik (1997), and Palik (1997) (compare Fig. 3).

      Information on aerosol size distribution as well as their shapes is also needed. In this study, we assume that the shape of aerosol is a sphere with a single-mode lognormal size distribution. The infrared spectra do not contain enough information to get real shape information. The width of size distribution of aerosol is assumed to be 0.2 and the effective radius ($R_{eff}$) is

set from 0.1 to 1 μm. Based on that, aerosol optical properties are calculated using the Lorenz-Mie theory. The code for this calculation has been developed by Mishchenko et al. (1999).





### 3.2.2 Artificial Spectra from LBLDIS

When considering down-welling emission from the atmosphere on a clear day, the main contribution to emission in the thermal infrared band are from the greenhouse gases, i.e. $CO_2$, $H_2O$, $N_2O$, $CO$, $CH_4$ and $O_3$. If there is a layer of cloud or aerosol,

the broad band emissions from cloud and aerosol can be observed as well. In order to model the spectrum, two radiative transfer models are used to simulate the down-welling emission from the atmosphere, one is the Line-by-Line Radiative Rransfer Model (LBLRTM, Clough et al. (2005)) for the gaseous contribution, another is the DIScrete Ordinate Radiative Transfer model (DISORT, Stamnes et al. (1988)) for calculation of scattering of the radiation on water droplets and aerosol particles. The coupled model is called LBLDIS and is used in several retrieval algorithms, such as MIXCRA (Turner, 2005),

and CLARRA (Rowe et al., 2013), and TCWret (Richter et al., 2022). Note, all this retrieval algorithms share the same forward models. The differences are the particular implementation, e.g. of the scattering.

Several thermal infrared emission spectra from the LBLDIS model are shown in Fig. 4. Under the same number density, different aerosol types exhibit unique characteristics in the infrared emission spectra, shown in Fig. 4a. Among those aerosols, the radiance emitted from sea salt is lowest, due to its small particles compared to other aerosols (Fig. 4b). Figure 4c shows

the thermal infrared emission spectra of atmosphere gases (clear sky) and different aerosols within atmosphere. According to Fig. 4c, there is no aerosol signal in some wavebands due to the domination of the gas emission, e.g. $CO_2$ in 640 - 690 $\mathrm{cm}^{-1}$ and $O_3$ in 1000 - 1100 $\mathrm{cm}^{-1}$. Apart from those wavebands, the four aerosol signals are obvious especially in 500 - 600 $\mathrm{cm}^{-1}$, 800 - 1000 $\mathrm{cm}^{-1}$, and 1100 - 1200 $\mathrm{cm}^{-1}$, which are selected as retrieval micro-windows (vertical lines in Fig. 4c). To make the signatures of aerosols more obvious, the spectra in Fig. 4d are shown in the form of the difference between the aerosol

and clear sky in those micro-windows . Based on that, the emission spectra in aerosol events are different from each other and independent, which means the emission from aerosols can be measured and aerosol types could be retrieved using emission FTS. The reason for avoiding the gas emissions is the dependency of the gas emissions on the temperature distribution in the atmosphere.

### 3.2.3 Error estimation

In order to investigate the precision of the retrieved values, artificial spectra simulated from LBLDIS are used to explore the performance of TCWret-V2 in the retrieval of aerosol types. Artificial spectra with preset values of AOD as well as $R_{\mathrm{eff}}$ are created using LBLDIS and then act as measured spectra retrieved by the algorithm. Specifically, we assume that all particles are concentrated on a single level, 2000 m above surface ground. The AOD's of sea salt, sulfate, dust and BC are set 0.1, respectively, with $R_{\mathrm{eff}}$ of 0.7 µm. The retrieval results suffer from several uncertainty sources:

– uncertainty of the aerosol height, which is similar to the error of aerosol layer temperature. In this study, the aerosol height is given by Lidar measurement.

– uncertainty of the humidity profile has a significant signal on the far-infrared emission spectrum, about 1500 - 2000 $\mathrm{cm}^{-1}$. Thus, the uncertainty of water vapor profile could change the radiance of emission spectrum, which might affect





the results of retrieval. In the retrieval processes, ERA5 hourly data on pressure levels from 1959 to present is used into retrieval (Hersbach et al., 2018).

– Calculation uncertainty in measured spectra is also an important uncertainty in retrieval, which could be caused by non-perfect emission of blackbodies. In this study, the total power calibration method (Revercomb et al., 1988) is used to calibrate the spectra.

– Measurement uncertainty is cuased by the noise on the spectral measurements, a random noise due to fluctuations on the detectors. The noise on the spectrum is assumed to be white in space and time.

– Databases uncertainty could be caused by uncertainty of aerosol complex refractive index. Both the real and imaginary part could have an influence on the accuracy of aerosol scattering properties look-up tables, as we mentioned in Sec. 3.2.1.

The artificial spectra with modifications are performed according Tab. 2. Compared with preset values, one can then compute the difference between retrieved values with preset values by perturbing each parameter, as shown in Fig. 5.

In Fig. 5, the original case means without any modifications as we mentioned before, which are close to preset values. The difference of AOD retrieved in original case with preset values are less than 0.005 at 900 $\mathrm{cm}^{-1}$, meaning convincing results using this retrieval algorithm. Besides, among those modification cases, uncertainties in measurements and water vapor profiles have small aeffect on the retrieval. The most sensitive parameter is database error, caused by uncertainty of complex refractive index. A dcrease of 10% in the real part of the complex refractive index will cause about 7% positive errors in AOD of sulfate, dust and BC, except for AOD of sea salt, which shows 18% negative error. While the 10% decrease of imaginary part of complex refractive index will cause about 4% negative errors in AOD of sulfate, dust and BC and 1% negative error in AOD of sea salt. Following the databases errors, the second most important error is the calibration error, an offset by 1 $\mathrm{mWsr}^{-1}\mathrm{m}^{-2}\mathrm{cm}^{-1}$ cuases an error of about 4% overestimation in results. Temperature error is the third most important effect in the aerosol retrieval.

In conclusion, when aerosol is present in the atmosphere, the emission from aerosol can be measured by FTS. According to forward model simulations, different aerosol types show their own features and are independent of each other. Using artificial spectra with preset values, the retrieval results are consistent with preset values under several possible perturbing scenery. Therefore, it is reliable to do aerosol components retrieval using the TCWret-V2.

## 4 Results

### 4.1 Aerosol-only Retrieval

On the 10[th] of June 2020 was an aerosol event in Ny-Ålesund. This aerosol event is chosen as our aerosol-only case. Figure 6 presents the four different aerosol classes and cloud based on the Lidar classification method (compare Sec.2.3). During this day, aerosols are mainly distributed below 1500 m. From 7:00 to 11:00, the thickness of coarse aerosol layer (dense aerosol in Lidar classification method in Sec. 2.3) near the surface decreases, while in the afternoon, this aerosol load increases and splits





into two layers, one near the surface and another, activated aerosol, appears at the height of about 500 m. Besides, there is a cloud signal around 8:00 at the height of 3500 m, which has been screened out in the aerosol-only retrieval in FTS.

Fig. 7 shows the result retrieved from FTS. From Fig. 7a, the dominant aerosol is sulfate above Ny-Ålesund, about AOD = 0.007 ± 0.0027 in daily average. The other three aerosols also exist, but in much lower AOD values compared with sulfate for most of the time, about AOD = 0.0039 ± 0.0029 for dust daily average, AOD = 0.0017 ± 0.0007 for BC and AOD = 0.0012

± 0.0002 for sea salt. From 9:00 to 11:00, the AOD of sulfate decreases with time and becomes similar to others at 11:00. After that, it increases slowly from 12:00 to 14:00, about 0.0135 at 14:00. Besides, retrieval results also show that among the remaining three aerosols, Dust is dominant and the AOD of Dust increases slightly in the afternoon. From Fig. 7b, sulfate, Dust and BC are small, about 0.3 μm, while the size of Sea Salt is larger, about 0.8 μm, which is likely to originate locally rather than transported over long distances.

From the measurement of sun-photometer, as shown in Fig. A1, the AOD of aerosol decreases from 8:00 to 11:00 and increases after 14:24. In Fig. 7a, the total AOD in FTS also shows similar daily variation and is mainly caused by daily changes of sulfate. According to MERRA-2 reanalysis data, as shown in Fig. A2, the daily variation of AOD on the 10th of June 2020 is mainly caused by sulfate and sea salt. Apart from sea salt, which shows limited signal in infrared waveband, the daily variation of sulfate in MERRA-2 is also consistent with FTS measurement. The good agreement of FTS measurements with

sun-photometer and MERRA-2 reanalysis data shows the good quality in the retrieval results of FTS.

In the afternoon, from Lidar measurement, aerosols become activated at the height of about 500 m. Since in FTS retrieval algorithm, the databases of aerosol do not include the liquid water, which means only dry particles are considered in our retrieval. The appearance of activated aerosol signal indicates that hygroscopic growth of aerosol should be considered in aerosol scattering properties look-up tables, which will be established in the future.

**4.2   Cloud-only Retrieval**

As we mentioned in Sec. 3.2, the intensity of emission spectra from thin clouds are very similar with those of aerosols. For showing the importance of Lidar measurement in the retrieval of FTS, a thin cloud-only case is selected and retrieved using two versions of retrieval algorithm.

Since aerosols are more or less present in the air, it is relatively hard to find a strictly cloud-only case. According to Lidar

measurement on the 11th of June 2020, there was a thin cloud in Ny-Ålesund, as shown in Fig. 8. Limited by the number of observations in Lidar, we only get four Lidar measurements. Compared with cloud or other aerosol classes, we assume that spherical aerosol shows the weakest signal in FTS. Based on that, the time period when only thin clouds exists and aerosols are relatively negligible is about 10:22. Then, both TCWret V1 and V2 are used for gaining cloud parameters and aerosol parameters respectively to show the importance of prior Lidar classification information in retrieval.

Tab. 3 shows the results of cloud parameters using Tcwret V1 and aerosol parameters using Tcwret V2. From the cloud retrieval, there is a piece of ice cloud ($\tau$ =0.06433) on the 11th of June 2020. While from the aerosol retrieval, the signal of this thin ice cloud is misunderstood and retrieved as four aerosols. Without information from Lidar measurement, both retrievals





are plausible. However, considering the Lidar as a reference, only cloud retrieval should be adopted, which means that the existence of clouds will interfere with the inversion of aerosols.

## 5  Conclusions

An FTS instrument, NYAEM-FTS, for measuring down-welling emitted radiation is operated since 2019. Combining with the Raman-Lidar KARL, the aerosols can be observed more comprehensively than by either instrument alone.

For FTS emission measurements, according to forward model simulation, the aerosol signatures of different aerosol types in the infrared wavelength are quite clear and independent. The retrieval algorithm TCWret-V1 (Richter et al., 2022) has been modified for retrieval of optical depth and $R_{eff}$ of different aerosol types. Combined Lidar and FTS, a scheme of instruments joint measurement is designed and applied to do aerosol component retrieval. The measurements from both instruments in two case studies are analyzed on 10[th] of June 2020 (aerosol-only case) and on 11[th] of June 2020 (cloud-only case) to show the potential synergy.

In the aerosol-only case study, 10[th] of June 2020, the signal of cloud and aerosols could be distinguished clearly using measurements from the Lidar KARL. From the emission FTS measurement, the sulfate is the dominant aerosol during the whole day. Comparing with sun-photometer, the daily variation of aerosol AOD is mainly effected by sulfate in infrared waveband. Comparing the results from NYAEM-FTS with MERRA-2 reanalysis data, the proportions of sulfate, dust and BC also show good agreement.

In the cloud-only case study, 11[th] of June 2020, Lidar could show the cloud or aerosol accurately and sensitively, giving a very good information on the state of the atmosphere. Without information from Lidar measurement, the signal of this thin ice cloud is retrieved as four aerosols from NYAEM-FTS measurements, which shows that the combination of both measurements is necessary.

The database used in TCWret-V2 does not include wet particles, which will be subject of a future study.

*Code and data availability.*  The latest version of TCWret can be downloaded from GitHub (https://github.com/RichterIUP/Total-Cloud-Water-retrieval). The Lidar data, spectra measured from the Emission FTS and retrieval resutls are available from the corresponding author upon request.

## Appendix A:  Aerosol Optical Thickness in AERONET and MERRA-2

The AERONET (AErosol RObotic NETwork) project is a federation of ground-based remote sensing aerosol networks. It is widely used as a ground-based reference for validation of aerosol retrievals. In Ny-Ålesund, a sun-photometer measuring solar extinction at several wavelengths is adopted to show the daily variance of AOD on 10[th] of June 2020, as shown in Fig. A1.

MERRA-2 (Modern-Era Retrospective analysis for Research and Applications version 2) is the latest version of global atmospheric reanalysis for the satellite era produced by NASA Global Modeling and Assimilation Office (GMAO) using the





Goddard Earth Observing System Model (GEOS) version 5.12.4. M2T1NXAER is an hourly time-averaged 2-dimensional data collection in MERRA-2. This collection consists of assimilated aerosol diagnostics, such as column mass density of aerosol

components (black carbon, dust, sea salt, sulfate, and organic carbon), surface mass concentration of aerosol components, and total extinction (and scattering) AOD at 550 nm. The dataset covers the period of 1980 to present. Fig. A2 shows the AOD of sea salt, sulfate, dust and BC in Ny-Ålesund on $10^{th}$ of June 2020.

## Appendix B:  Theory of Modified TCWret

The retrieval method adopted in modified TCWret for aerosol case is the optimal estimation method (Rodgers, 2000), the
relationship between a measured emission spectrum $y$ and unknown aerosol state $x$ can be described by a simple mathematical model, as follows:

$$y = F(x) + \varepsilon \tag{B1}$$

where $F(x)$ is the forward model and $\varepsilon$ is the error of observation. The solution of the inverse problem is the state $x$ minimizing a cost function $\xi^2(x)$ usually defined as:

$$\xi^2(x) = [y - F(x)]^T S_y^{-1} [y - F(x)] + [x_a - x] S_a^{-1} [x_a - x] \tag{B2}$$

where $S_y^{-1}$ is the inverse measurement error covariance matrix, containing the variances of the spectral radiance; $x_a$ is the apriori; $S_a^{-1}$ is the inverse error of the a priori covariance matrix $x_a$; The state vector $x$ in modified TCWret is defined as follows: $x = (\tau_{seasalt}, \tau_{sulfate}, \tau_{dust}, \tau_{BC}, r_{seasalt}, r_{sulfate}, r_{dust}, r_{BC})$ $\tau$ means AOD of aerosols, and $r$ means $R_{eff}$ of aerosols.

Since the forward model is a non-linear function, which means an iterative method is needed to minimize the cost function $\xi^2(x)$, given as follow:

$$x_{n+1} = x_n + s_n \tag{B3}$$

Here $x_n$ and $x_{n+1}$ are the aerosol parameters of the $n-th$ and $(n+1)-th$ step, and $s_n$ is the modification of the aerosol parameters during the $n-th$ iteration. For weak non-linear problems, the Gauss-Newton (GN) method can be successfully
applied, while in significant non-linear situations, the GN method is not guaranteed to decrease the cost function, therefore the steepest descent could be used. The Levenberg-Marquardt method modification combines both methods by starting with the deepest descent method far away from the minimum and using the GN method near the minimum. At each iteration, a damping factor $\mu$ is adjusted in such a way that if the step results in a decrease in the cost-function, the damping factor $\mu$ is decreased, bringing the next step closer to the GN step. If the step causes the cost function to increase, the iteration is repeated
with a higher damping factor $\mu$, resulting in a step closer to the gradient descent direction Ceccherini and Ridolfi (2010). The adjustment vector $s_n$ could be determined by the governing equation, as follows:

$$(K_n^T S_y^{-1} K_n + S_a^{-1} + \mu^2 S_a^{-1}) s_n = K_n^T S_y^{-1} [y - F(x_n)] + S_a^{-1} \cdot (x_a - x_n) \tag{B4}$$





$K = (\frac{\partial F(x_i)_n}{\partial x_i})$ is the jacobian matrix, $i$ means parameters in the state vactor; $S_y^{-1} = diag(\sigma_i^{-1})$ is the inverse measurement error covariance matrix, containing the variances of the spectral radiance; $x_a$ is the apriori; $S_a^{-1}$ is the inverse error of the a priori covariance matrix $x_a$; $\mu^2 \cdot S_a^{-1}$ is the Levenberg-Marquardt (LM) term, as we mentioned before. $F(x_i)$ is the calculated spectral radiance and $y$ is the measured spectral radiance.

The iteration is said to have converged, if the cost function $\xi^2$ does not change anymore, i.e. the change in the cost function $\xi$ is below a threshold. This threshold is set 0.001 in this study, i.e. the iteration has converged if

$$\frac{\xi^2(x_{n+1}) - \xi^2(x_{n+1})}{\xi^2(x_n)} < 0.001 \tag{B5}$$

## B0.1   Averaging Kernels

The averaging kernels are a useful diagnostic tool to characterize the solution of the retrieval. In TCWret, averaging kernels are calculated via

$$A = \frac{\partial x_r}{\partial x} = \frac{\partial x_r}{\partial y}\frac{\partial y}{\partial x} = T_r \cdot K_r \tag{B6}$$

where $x_r$ is the retrieved state vector; $x$ is the true value of state vector; $T_r$ is the final transfer matrix $T$ and $K_r$ is the final jacobian matrix. According to Ceccherini and Ridolfi (2010), the final transfer matrix could be calculated as follows:

$$\begin{cases} T_0 = \mathbf{0} \\ T_{n+1} = G_n + (\mathbf{I} - G_n K_n - M_n S_a^{-1})T_n \\ G_n = M_n K_n^T S_y^{-1} \\ M_n = (K_n^T S_y^{-1} K_n + S_a^{-1} + \mu^2 D_n)^{-1} \end{cases} \tag{B7}$$

where $\mathbf{0}$ is a zero matrix and $\boldsymbol{I}$ is an identity matrix, other quantities are described before. The matrices $K_n$ are calculated in Eq. B4. The calculation of the transfer matrix is performed in parallel to the minimisation.

*Author contributions.* PR implemented TCWret and DJ developed it into the retrieval of aerosol parameters using TCWret. MP designed and built the measurement setup, performed measurements and gave advice in the development of TCWret. CR performed Lidar measurements and gave advice in using the Lidar data and the sun-photometer data. XS gave the advice in forward model simulation. MB gave the advice of calibration processes. JN gave advice in the setup of the measurement and the development of TCWret. All authors contributed to manuscript revisions.

*Competing interests.* The authors declare no competing interests



*Acknowledgements.*  We gratefully acknowledge funding from the Transregional Collaborative Research Centre TR 172 – Arctic Amplification: Climate Relevant Atmospheric and Surface Processes ((AC)$^3$), project E02: Ny-Ålesund Column Thermodynamic Structure, Clouds, Aerosols, Trace Gases and Radiative Effects.



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



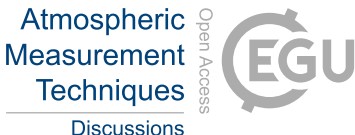

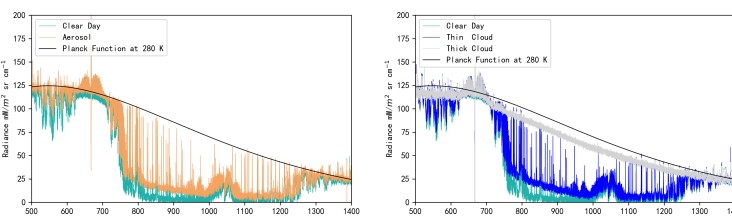

**Figure 1.** Four different emission spectra measured by NYAEM-FTS in clear sky (green), aerosol (yellow, left), thin cloud (blue, right) and thick cloud (gray, right) event respectively. The radiance calculated using Planck function at 280 K (black line) is presented in this figure. Note: the emission around $650\,\mathrm{cm}^{-1}$ is $CO_2$ in the laboratory.



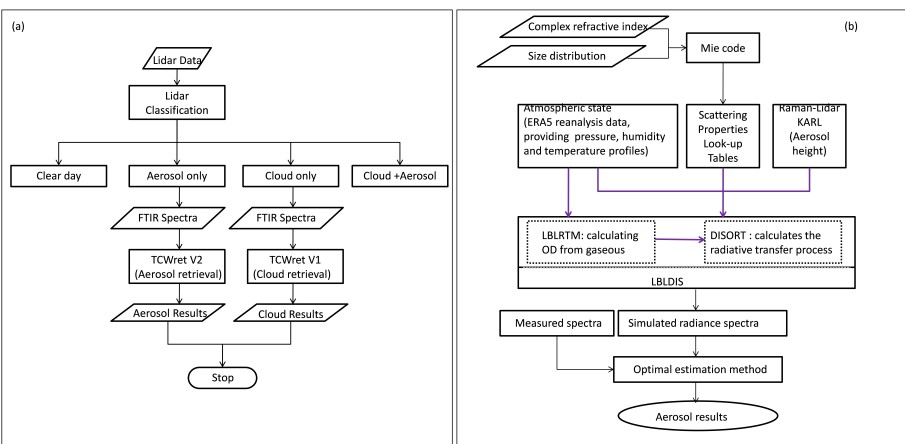

**Figure 2.** Instruments Joint Observation Scheme (a) and flow diagram of TCWret-V2 (b).



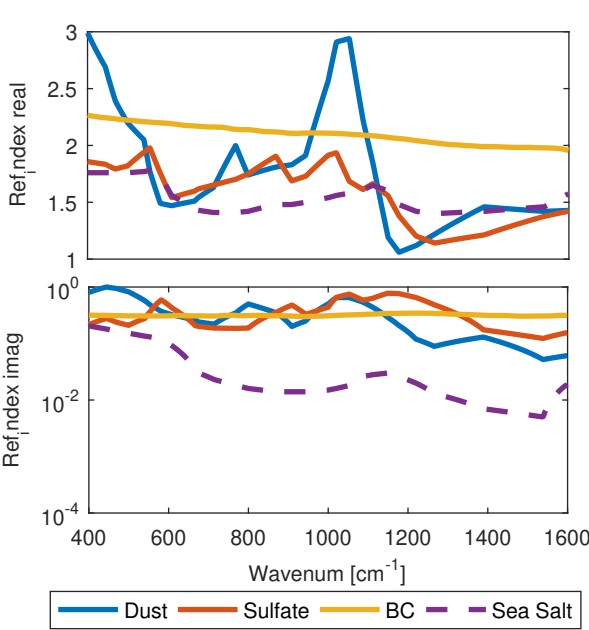

**Figure 3.** The complex refractive index of dust, sulfate, BC, and sea salt. The complex imaginary refractive index of sulfate and dust are based on OPAC/GADA database, BC from Chang and Charalampopoulos (1990), sea salt from Eldridge and Palik (1997), and Palik (1997). Those data have been downloaded from: http://eodg.atm.ox.ac.uk/ARIA/.



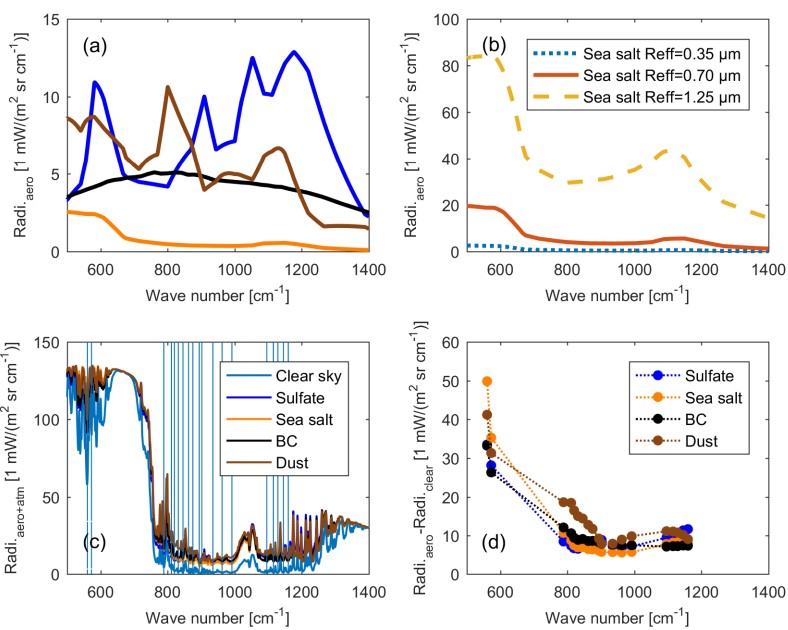

**Figure 4.** The emission spectra of small aerosol particles (dust in brown, sulfate in blue, sea salt in orange, BC in black) with Reff = 0.35 μm and number density = 2000 $cm^{-3}$ (a); The emission spectra of sea salt with different particle sizes (b); The emission spectra of aerosols ($AOD_{900\,cm^{-1}}$ = 0.1) with atmosphere gases and clear sky case (c); The difference between total emission spectra of aerosol and clear sky case in micro windows (d). The vertical blue lines in (c) show the mid-values of micro windows selected for retrieval. The emission spectra are simulated from LBLDIS with the resolution of 1 $cm^{-1}$.





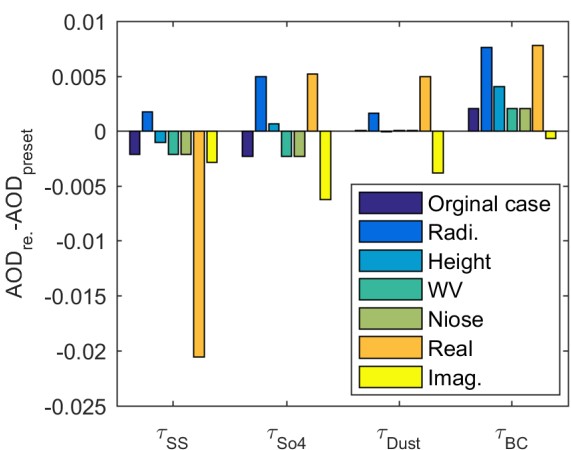

**Figure 5.** The difference between retrieved AOD in orginal case and several possible perturbing scenery (compare Table 2.) with preset values.





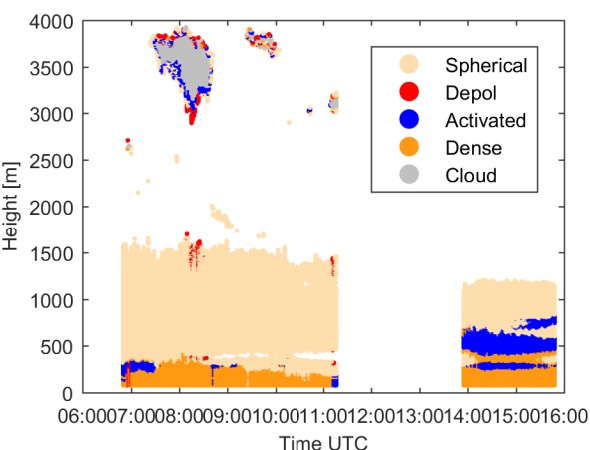

**Figure 6.** Four different aerosol classes (spherical particles in light yellow, depolarization particles in red, activated particles in blue, and dense particles in deep yellow) and cloud (gray) based on Lidar classification method on 10[th] of June 2020.



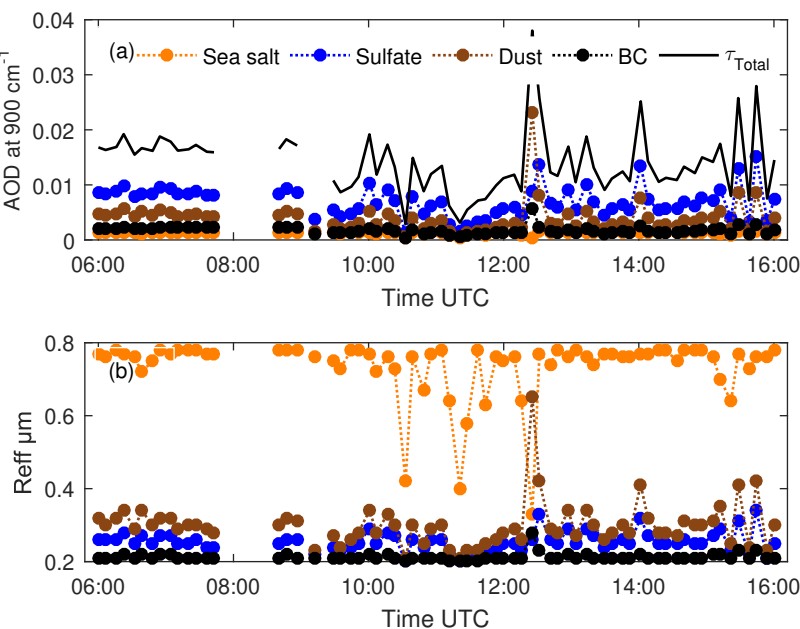

**Figure 7.** AOD of sea salt (orange), sulfate (blue), dust (brown), BC (black) and total AOD (black solid line) retrieved from emission FTS measurements (a) and Reff results with same color information (b) on 10[th] of June 2020.





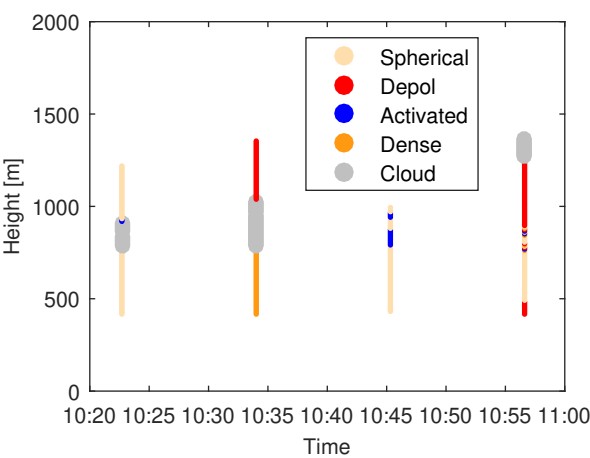

**Figure 8.** Four different aerosol classes (spherical particles in light yellow, depolarization particles in red, activated particles in blue, and dense particles in deep yellow) and cloud (gray) based on Lidar classification method on 11[th] of June 2020.



**Table 1.** Aerosol classification by Lidar measurements as give by Ritter et al. (2016)

| Classes | $\beta_{532}^{aer}(Mm^{-1}sr^{-1})$ | $\delta_{532}^{aer}$ | CR | Description |
|---|---|---|---|---|
| Clear day | $\beta < 0.4$ | $\delta < 2.05\%$ | | Clear day |
| Clear depol. | $\beta < 0.4$ | $\delta \geq 2.05\%$ | | Clear day with polarized signal. |
| Spherical Aerosol | $0.4 \leq \beta < 1$ | $\delta < 2.05\%$ | | Spherical fine particles, possibly from long-distance transportation, e.g. sulfate. |
| Depol. Aerosol | $0.4 \leq \beta < 1$ | $\delta \geq 2.05\%$ | | Polarized fine particles with irregular shapes, e.g. dust. |
| Activated Aerosol | $1 \leq \beta \leq 3$ | $\delta < 2.05\%$ | $CR < 1.7$ | Aerosol hygroscopic growth into larger size, e.g. sea salt, sulfates. |
| Dense Aerosol | $1 \leq \beta \leq 3$ | | $CR \geq 1.7$ | Medium size aerosol, e.g. sea salt, dust. |
| Cloud | $\beta > 3$ | | | Cloud |



**Table 2.** Parameter errors and modifications in artificial spectra

| Parameters | Modifications |
| --- | --- |
| Height of aerosol | +10% (200 m) |
| Water vapor profiles | -10% |
| Calibration error | +1 mW/sr*m2*cm-1 |
| Measurement error | Normally distributed noise with mean value of 0 and variance of 1 |
| Complex refractive index (real part) | -10% |
| Complex refractive index (imaginary part) | -10% |



**Table 3.** Cloud parameters and aerosol parameters using TCWret-V1 and TCWret-V2 on 11[th] of June 2020

| Tcwret version | $\tau_{Liquidcloud}$ | $\tau_{Icecloud}$ | $\tau_{Seasalt}$ | $\tau_{Sulfate}$ | $\tau_{Dust}$ | $\tau_{BC}$ |
|---|---|---|---|---|---|---|
| Tcwret V1 | 0 | 0.0643 | | | | |
| Tcwret V2 | | | 0.000276 | 0.000425 | 0.000208 | 0.000219 |



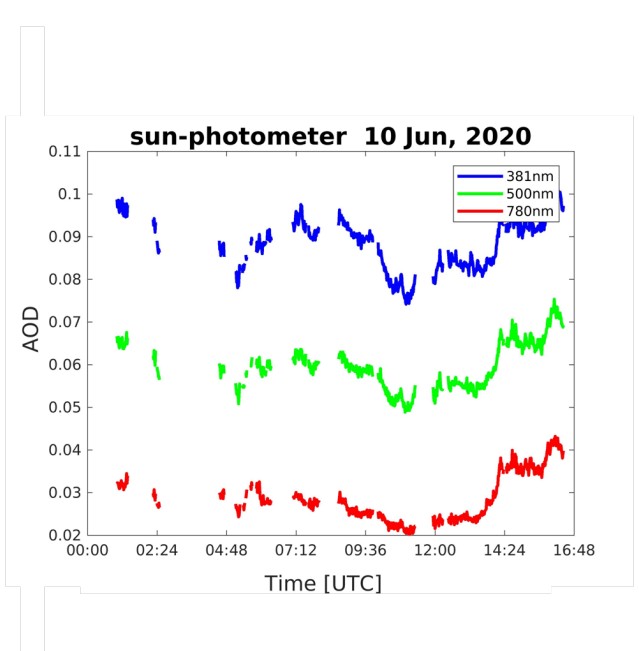

**Figure A1.** Sun-photometer aerosol optical depth in Ny-Ålesund on 10[th] of June 2020. From: https://aeronet.gsfc.nasa.gov/.



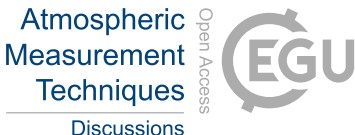

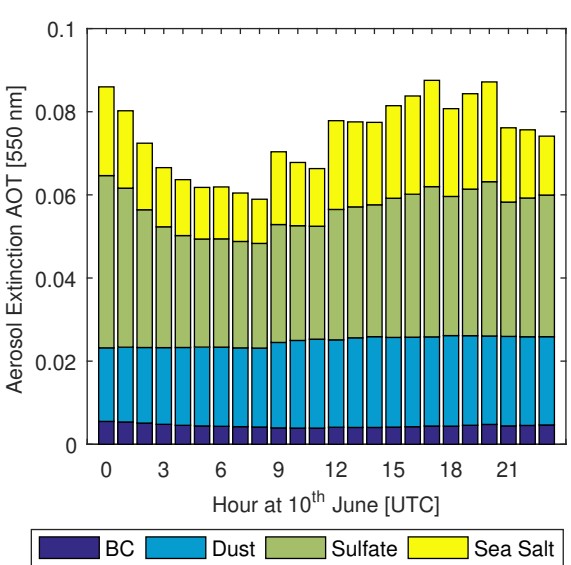

**Figure A2.** Merra-2 reanalysis aerosol data in Ny-Ålesund on 10th of June 2020.

From: https://goldsmr4.gesdisc.eosdis.nasa.gov/data/MERRA2/M2T1NXAER.5.12.4/.