# Peer review of "Ground-based remote sensing of aerosol properties using high resolution infrared emission and Lidar observations in the high Arctic"

_Atmospheric Measurement Techniques, 2022_

## Author Comment (AC1)

R1

The authors thank the reviewer for the careful review and the hints to improve the manuscript.

**L53-54: If there are any previous studies regarding to the remote sensing of aerosols using the FTS, you should mention them.**

Answer:

Turner (2008) and Rathke et. al. (2002) are added to the new version of the manuscript.

**L144-145: Sea salt and dust particles can have larger radius than 1.0 μm. Why is the maximum radius 1.0 μm?**

Answer:

The main reason for setting the upper limit of the Reff to 1 μm is that aerosols in the Arctic region is often below 1 μm, according to the measurements of aerosol size distribution in the Arctic area ((Asmi et al., 2016; Park et al., 2020; Boyer et al., 2022)). In addition, if we do not make such constrain into 1 μm, sometimes, the retrieval of fine particles, such as sulfate and BC, will be mathematically increased for a better fit of the spectrum, which is artificial. Since that, we decided to make such constraint.

When the retrieved Reff of sea salt is close to 1 μm and sea salt is dominant during a aerosol event, we will extend the databases to 2.5 μm or even larger size and rerun the retrieval for a better result. While in this paper, the dominant one is sulfate, so the maximum radius is set 1 μm.

**L159: The result of sea salt in Fig. 4a is calculated using the same effective radius as the other particles. When the effective radius of the other particles is 0.70, and 1.25 μm, does the sea salt have comparable radiances with those of the other particles?**

Answer:

Other aerosol types with larger size are presented in Fig. A1. As a result, the radiance from sea salt with the same size as other aerosols is significantly lower; only when sea salt has a large particle size, whereas other particles are smaller, are both radiances comparable.

**The light absorbing capability of the particles could affect the radiances. Does the smallest light absorbing capability of sea salt affect the radiances?**

The smallest light absorbing capability of sea salt affects the radiance. The light absorbing capability is already included in the calculation (both in databases and LBLDIS).

[Figure]

Figure A1. The emission spectra of large aerosol particles (dust in brown, sulfate in blue, sea salt in orange, BC in black) with Reff = 0.70 μm and number density = 2000 cm$^{-3}$

**L181: Calculation ---> Calibration**

Answer:

Corrected.

**L203: Can you show the reliable range of the retrieval AOD from the results in Fig. 5?**

Answer:

Aerosol AOD is reliable when it is greater than 0.003. In order to show the reliable range of the retrieval AOD, using similar artificial spectra but for several setting of AOD from 0.001 to 0.1 at 900 cm$^{-1}$, the relative uncertainties of AOD in original cases with several preset values are given in Fig. 5b. It shows that the reliability range of the AOD retrieval is reliable for and AOD > 0.003.

[Figure]

Figure 5. The difference between retrieved AOD in original case and several possible perturbing scenery (compare Table 2.) with preset values (a). The difference between retrieved effective radius in original case and several possible perturbing scenery (compare Table 2.) with preset values (c). The relative uncertainties of AOD in original cases with several preset values (b). The relative uncertainties of effective radius in original cases with several preset values (d).

Note: Fig. 5 (a) and (c) share the same legend in (a); Fig. 5 (b) and (d) share the same legend in (b). The artificial spectra with several preset values in (b) and (d) means the AOD's of aerosols are set from 0.001 to 0.1, compare Line 173. With the same method but for different preset AODs, the reliable range of AOD are given in (b) and (d).

**L215: In general, AOD of BC would be lowest in the four types. The AOD of BC in the MERRA-2 reanalysis is smallest in the four particles. However, the AOD of BC in the retrieval results is larger than AOD of sea salt. The authors show that the AOD of sea salt is underestimated and the AOD of BC is overestimated in Fig. 5. Should we interpret that the AOD of sea salt is underestimated and the AOD of BC is overestimated in Fig. 7?**

Answer:

As we mentioned in former answer, the AOD is considered reliable if larger than 0.003. In Fig.7, both BC and sea salt are close to 0.003. Sulfate and dust are much higher than BC and sea salt. Therefore, the values of BC and sea salt are not valid, and are plotted only for informational purposes.

**L217-219: The authors do not show the uncertainties of the retrieved effective radius. The effective radii of sulfate, dust, and BC are 0.3 µm in Fig. 7.**

Answer:

Fig. 5c shows the similar information as Fig. 5a but for aerosol effective radius. It shows that only the calibration error and the imaginary part of complex refractive index will cause an error of aerosol effective radius, less than 0.02 µm. Fig. 5d shows the relative uncertainties of effective radius in original cases with several preset values same as Fig. 5b. The relative uncertainty of the effective radius retrieval is more stable than that of AOD, less than ± 5% in all artificial spectra cases.

**Does the a priori information affect these results?**

**Yes, errors in measurements are unavoidable and the information content of the measurement itself limited, which means that the a priori information affects the results.** The influence of a priori on the retrieval is encoded in the AVK matrix, telling us the information that how much the measurement is in the results and how much a priori. This is also the reason that we present the AVK in the appendix.

To make such effect more intuitively understood, several retrievals are conducted using the artificial spectrum in the original case in Fig. 5a by changing the apriori of aerosol effective radius from 0.3 to 0.7 µm. The result is shown in Fig. 6. For both AOD and aerosol effective radius retrieval, the more accurate the aprior information of aerosol effective radius is, the closer the retrieved result is to the preset value. Furthermore, sulfate and BC are more sensitive to the apriori information of aerosol effective radius than sea salt and dust, which means good apriori information about the size of sulfate and BC is helpful in the accuracy of retrieval.

[Figure]

Figure 6. The retrieved AOD and Reff using the artificial spectrum (all aerosols' AOD set 0.1 with Reff setting 0.7µm, compare original case in Fig. 5a) with different apriori of Reff from 0.3 to 0.7 µm.

**L275: Organic carbon (OC) is one of the major components in the tropospheric aerosols.**

**The authors do not consider OC in the retrieval method, but actually OC would affect the retrieval results. What is the influence of OC?**

Answer:

Organic carbon (OC) is one of the major components in the tropospheric aerosols. It is not considered because there are no data in the complex refractive index at infrared waveband of OC. There are many types of OC, each of them may have a different spectral signature. However if there are specral features which are not fitted, e.g. due to the presence of aerosol types not accounted for in the scattering database, the error margin on the retrieved aerosol types will be increased.

**L282-319: Matrices are printed in boldface, and vectors in boldface italics. See "submission" of AMT web page.**

Answer:

Corrected.

**L310: Averaging kernels are described in only this section. Were the averaging kernels used in this study?**

Answer:

Averaging kernels are given but not used in this study, since we haven't use AVK to do data comparison. However the averaging kernels belong to the retrieved result, because they include much information about the retrieval results, like the how influence is exerted by the a priori and how independent the retrieved quantities are from each other.

**Reference:**

Turner, D. D.: Ground-based infrared retrievals of optical depth, effective radius, and composition of airborne mineral dust above the Sahel, J. Geophys. Res.: Atmos., 113, https://doi.org/https://doi.org/10.1029/2008JD010054, 2008.

Rathke, C., Notholt, J., Fischer, J., and Herber, A.: Properties of coastal Antarctic aerosol from combined FTIR spectrometer and sun photometer measurements, Geophys. Res. Lett., 29, 46–1, 2002.

Asmi, E., Kondratyev, V., Brus, D., Laurila, T., Lihavainen, H., Backman, J., Vakkari, V., Aurela, M., Hatakka, J., Viisanen, Y., Uttal, T., Ivakhov, V., and Makshtas, A.: Aerosol size distribution seasonal characteristics measured in Tiksi, Russian Arctic, Atmos. Chem. Phys., 16, 1271–1287, https://doi.org/10.5194/acp-16-1271-2016, 2016.

Park, J., Dall'Osto, M., Park, K., Gim, Y., Kang, H. J., Jang, E., Park, K.-T., Park, M., Yum, S. S., Jung, J., Lee, B. Y., and Yoon, Y. J.: Shipborne observations reveal contrasting Arctic marine, Arctic terrestrial and Pacific marine aerosol properties, Atmos. Chem. Phys., 20, 5573–5590, https://doi.org/10.5194/acp-20-5573-2020, 2020.

Boyer, M., Aliaga, D., Pernov, J. B., Angot, H., Quéléver, L. L. J., Dada, L., Heutte, B., Dall'Osto, M., Beddows, D. C. S., Brasseur, Z., Beck, I., Bucci, S., Duetsch, M., Stohl, A., Laurila, T., Asmi, E., Massling, A., Thomas, D. C., Nøjgaard, J. K., Chan, T., Sharma, S., Tunved, P., Krejci, R., Hansson, H. C., Kulmala, M., Petäjä, T., Sipilä, M., Schmale, J., and Jokinen, T.: A full year of aerosol size distribution data from the central Arctic under an extreme positive Arctic Oscillation: Insights from the MOSAiC expedition, Atmos. Chem. Phys. Discuss., 2022, 1–45, https://doi.org/10.5194/acp-2022-591, 2022.

---

## Author Comment (AC2)

R2

**The author are targeting an important topic, the temporal change and retrieval of different aerosol classes in the Arctic. This is one of the topics which have to be determined and discussed in the context of the phenomenom of Arctic amplification and climate warming. They are using the information and synergy of different intrument types, present a joint observation scheme and a flow diagram of the aerosol retrieval scheme.The aerosol classes were determined for two different atmospheric conditions, a cloud case and a cloud free case. Most of the work is done and hidden behind those two schemes and this and the calculation of results and error bars could be described in more detail. Unfortunately the examination is limited to two single example days. A temperal change of the aerosol classes for a longer time period would be much more comprehensive. The results are discussed and a daily cycle is presented for different aerosol classes. Overall the manuscript is well written and after some minor changes almost ready for publication.**

Answer:

Thank you very much.

We agree, that the evaluation of only two days does not contribute much information to the problem of the Arctic amplification. Hence, an analysis of the whole time series is in preparation. The present study is designed to discuss the method to obtain aerosol data from the measurements, and give a characterization of the result.

---

## Author Comment (AC3)

R3

The authors thank the reviewer for the careful review and the hints to improve the manuscript.

**Q1: ... I found a little poor the aerosol characterization and I recommend to the authors to add supplementary information from AERONET for a major aerosol characterization, taking into account that only two days are characterized. More detailed information will benefit the section of results. The use of SDA algorithm to discriminate coarse and fine particles can be useful. Furthermore, the data allows to combine visible and far infrared information.**

Answer:

Using emission FTS to do aerosol component retrieval in infrared waveband could help to obtain more information about the effect of aerosols on the Arctic atmosphere. We agree that it is advisable and worthwhile to combine AERONET measurements and the method outlined here. Such studies are planned for the future together with an evaluation if the lidar data from cloudnet can be used to replace the data of the KARL lidar.

**Q2: The paper is not focused in the methodology, already described elsewhere (Richter et al, 2020), hence this section may be shortened, for example the equations may be removed and give a more qualitative description and problems involved. On the contrary, an idea more quantitative of the associated errors to the retrieved parameters, as AOD, would be appreciated.**

Answer:

Richter et.al. 2020 used emission spectra obtained by an FTIR instrument to infer cloud parameters, not aerosol composition. The reviewer is correct, the general method is the same, but the scattering tables are different and so is the characterization. This paper is supposed to serve as a base for further studies.

**Q3: The Appendix A may be included in the Section 3**

Answer:

Appendix A has been corrected in the Section 3.3.

Specify comments

**Line 135, modify the word DSIORT, Stamnes et al. (1988))**

Answer:

Corrected.

**Line 222, Capital letter are used along the document for TCWret V1 and V2: modify Tcwret V1 and aerosol parameters using Tcwret V2.**

Answer:

Corrected.

---

## Author Response (AR1)

**Point to Point corrections in the paper.**

Section changes:

1.  All three reviewers want more retrieved data from FTS, therefore, in section 4.3, we add long time period observation.

2.  Cloud-only case is deleted. We want to prove Lidar information is important in distinguishing aerosol or cloud, which is already proved in the aerosol case.

3.  The results of artificial spectra used as measured spectra are moved into section 4.1, which is in section 3.2.3 in the former version of the manuscript.

Details changes according to three reviewers:

Turner (2008) and Rathke et. al. (2002) are added.
*(R1 If there are any previous studies regarding to the remote sensing of aerosols using the FTS, you should mention them.)*

The main reason for setting the upper limit of the Reff to 1 \unit{\mu m} is that aerosols in the Arctic region are often below 1μm, according to the measurements of aerosol size distribution in the Arctic area ((Asmi et al., 2016; Park et al., 2020; Boyer et al., 2022)).   In addition, if the such constraint is not set to 1 \μm, occasionally, the retrieval of fine particles, such as sulfate and BC, will be mathematically increased for a better fit of the spectrum, which is artificial. Because sea salt can be larger than 1μm, when the retrieved   Reff of sea salt is close to 1μm and sea salt is the dominant aerosol, the database of sea salt is extended to 2.5μm and the retrieval is run again.
  *(R1 Sea salt and dust particles can have a larger radius than 1.0 μm. Why is the maximum radius 1.0 μm?)*

Other aerosol types with larger sizes are presented in Fig. A1. As a result, the radiance from sea salt with the same size as other aerosols is significantly lower; only when sea salt has a large particle size, whereas other particles are smaller, are both radiances comparable.
*(R1 The result of sea salt in Fig. 4a is calculated using the same effective radius as the other particles. When the effective radius of the other particles is 0.70, and 1.25 μm, does the sea salt have comparable radiances with those of the other particles?)*

Section 3.3 AOD in AERONET and MERRA-2 are added.
*(R3: Appendix A may be included in Section 3)*

In order to show the reliable range of the retrieval AOD, using similar artificial spectra but for several settings of AOD from 0.001 to 0.1 at 900 cm$^{-1}$, the relative uncertainties of AOD in original cases with several preset values are given in In Fig. 5b. It shows that when the AOD of the aerosol is low, less than 0.003, the retrieval result is distorted. However, as the AOD of aerosol

gradually increases, the aerosol signals in the infrared waveband become more recognizable, the retrieval result of aerosol gradually approaches the true value, becoming more reliable. Therefore, we propose that when the AOD is greater than 0.003, the retrieval results are reliable.

*(R1 Can you show the reliable range of the retrieval AOD from the results in Fig. 5?)*

Organic carbon (OC) is one of the major components in the tropospheric aerosols. It is not considered because there are no data in the complex refractive index at infrared waveband of OC. There are many types of OC, each of them may have a different spectral signature. However if there are spectral features which are not fitted, e.g. due to the presence of aerosol types not accounted for in the scattering database, the error margin on the retrieved aerosol types will be increased.

*(R1 Organic carbon (OC) is one of the major components in tropospheric aerosols. The authors do not consider OC in the retrieval method, but actually, OC would affect the retrieval results. What is the influence of OC?)*

The averaging kernels belong to the retrieved result, because they include much information about the retrieval results, e.g. how much influence is exerted by the a priori and how independent the retrieved quantities are from each other. On the diagonal elements one finds the derivatives of each element in the retrieved state vector with respect to its corresponding element in the true state vector. From the averaging kernel, the AOD of sulfate is the parameter least dependent on a priori information, followed by the AOD of dust and sea salt. Except for dust, all other aerosol size information is difficult to be retrieved.

*(R1 Does AVK used in this study?)*